# On the Possible Effect of Phytic Acid (Myo-Inositol Hexaphosphoric Acid, IP6) on Cytochromes P450 and Systems of Xenobiotic Metabolism in Different Hepatic Models

**DOI:** 10.3390/ijms25073610

**Published:** 2024-03-23

**Authors:** Veronika Frybortova, Stefan Satka, Lenka Jourova, Iveta Zapletalova, Martin Srejber, Philippe Briolotti, Martine Daujat-Chavanieu, Sabine Gerbal-Chaloin, Pavel Anzenbacher, Michal Otyepka, Eva Anzenbacherova

**Affiliations:** 1Department of Medical Chemistry and Biochemistry, Faculty of Medicine and Dentistry, Palacky University Olomouc, 775 15 Olomouc, Czech Republic; veronika.frybortova@upol.cz (V.F.); eva.anzenbacherova@upol.cz (E.A.); 2Department of Pharmacology, Faculty of Medicine and Dentistry, Palacky University Olomouc, 775 15 Olomouc, Czech Republic; iveta.zapletalova@upol.cz; 3Czech Advanced Technology and Research Institute (CATRIN), Palacky University Olomouc, 779 00 Olomouc, Czech Republic; 4Institute for Regenerative Medicine and Biotherapy (IRMB), University Montpellier, INSERM, CHU Montpellier, F-34000 Montpellier, Francesabine.gerbal-chaloin@inserm.fr (S.G.-C.); 5IT4Innovations, VŠB—Technical University of Ostrava, 708 00 Ostrava, Czech Republic

**Keywords:** phytic acid, IP6, phytates, cytochrome P450, CYP1A, drug metabolism

## Abstract

As compounds of natural origin enter human body, it is necessary to investigate their possible interactions with the metabolism of drugs and xenobiotics in general, namely with the cytochrome P450 (CYP) system. Phytic acid (myo-inositol hexaphosphoric acid, IP6) is mainly present in plants but is also an endogenous compound present in mammalian cells and tissues. It has been shown to exhibit protective effect in many pathological conditions. For this paper, its interaction with CYPs was studied using human liver microsomes, primary human hepatocytes, the HepG2 cell line, and molecular docking. Docking experiments and absorption spectra demonstrated the weak ability of IP6 to interact in the heme active site of CYP1A. Molecular docking suggested that IP6 preferentially binds to the protein surface, whereas binding to the active site of CYP1A2 was found to be less probable. Subsequently, we investigated the ability of IP6 to modulate the metabolism of xenobiotics for both the mRNA expression and enzymatic activity of CYP1A enzymes. Our findings revealed that IP6 can slightly modulate the mRNA levels and enzyme activity of CYP1A. However, thanks to the relatively weak interactions of IP6 with CYPs, the chances of the mechanisms of clinically important drug–drug interactions involving IP6 are low.

## 1. Introduction

Phytic acid (myo-inositol hexaphosphoric acid, IP6) is the most abundant inositol phosphate in nature and is present in mammalian cells as well as in plants, mainly in cereals, legumes, nuts, and vegetables. IP6 strongly binds mineral cations such as Fe^2+^, Ca^2+^, Cu^2+^, and Zn^2+^; therefore, phytates, complexes with cations, are the most common forms of IP6 in nature [1]. Phytases (myo-inositol hexakisphosphate phosphohydrolases) catalyze the stepwise hydrolysis of IP6 to inorganic phosphate and myo-inositol via less-phosphorylated derivatives (IP5–IP1). Two groups of internationally accepted phytases are named after the site of the first hydrolysis: 3-phytases (EC 3.1.3.8) and 6-phytases (EC 3.1.3.26) [1]. Some studies have shown that monogastric animals like poultry, pigs, and also humans have very low intrinsic phytase activity [2,3]. For this reason, it was suggested that IP6 can affect the bioavailability of minerals in the gastrointestinal tract, and this is why it has been considered an antinutritional compound. On the other hand, IP6 is a natural compound with antioxidant properties which has beneficial effects in many pathological conditions, including neurodegenerative diseases, cardiovascular diseases, and cancer; likewise, a hepatoprotective effect has been observed [4,5]. The potential of IP6 to slow tumor growth has been shown in different cell types at 1–5 mM concentrations [6]. Also, IP6 is known to be a source of 5-diphospho inositol pentakisphosphate, formed by inositol hexakisphosphate kinase, playing a role in processes such as cytoskeletal remodeling, cellular migration, gene expression, and DNA repair and immunity [7]. 

Drug metabolism is usually a two-phase process that allows for the excretion of foreign substances due an increase in their polarity. The main organ where xenobiotic metabolism takes place is the liver, although the biotransformation of drugs occurs in many other organs and tissues. Cytochromes P450 (CYPs) are key enzymes in the first phase, being involved in the metabolism of 70–80% of clinically used drugs, and they contribute to enormous variability in drug response [8,9,10]. The activity of CYPs is affected by many factors, which can be divided into direct and indirect mechanisms of action. The direct mechanisms involve the binding of xenobiotics straight to the active or allosteric site of the enzyme, exerting effects on enzymatic function. On the other hand, mostly indirect mechanisms, such as transcriptional regulation, also influence the amount of active enzyme in the cell. The expression of CYPs, including CYP1A1 and CYP1A2, is affected by many factors, like gender, age, or even hypoxia, and CYPs are regulated by specific xenoreceptors such as the aryl hydrocarbon receptor (AhR); nuclear receptors such as pregnane X receptor, farnesoid X receptor, constitutive androstane receptor, liver X receptor, and retinoic acid receptor (RAR) [10,11]; and by post-transcriptional regulation by miRNA [12]. To date, many ligands of these receptors have been identified, and a variety of them are dietary components [13,14]. Diet-derived metabolites or xenobiotics that share a common pathway with clinically used drugs have the potential for harmful drug interactions [15]. Different dietary habits may contribute to interindividual differences in CYP activity, potentially leading to alterations in the pharmacokinetics of administered medicaments.

Examinations of the interaction of a substance with CYP usually begin with optical absorption spectroscopy studies of changes in the spectra of microsomal CYP due to addition of an examined substrate [16]. The spectral changes may indicate either a perturbation of electron density distribution in the heme active site by binding to the substrate binding site (pocket) or by direct binding to the heme iron. The characteristic difference spectra exhibit either maximum at about 394 nm (Type I) or 430 nm (Type II), while the third type of spectral changes, labeled as reverse Type I (also named modified Type II), exhibits maximum at about 415–435 nm, indicating different modes of interaction [17,18]. 

For this paper, the majority of our experiments were performed on CYPs 1A2 and 1A1 of the 1A subfamily. This subfamily consists of two enzymes, CYP1A2 (primarily hepatic) and CYP1A1 (mainly extrahepatic, expressed in the liver after induction), which have highly similar structures and overlapping substrate specificities (e.g., both share 7-ethoxyresorufin O-deethylation activity [19]). CYP1A1 and CYP1A2 are structurally close proteins, with 70% sequence identity in their coding gene regions [9]. Because of their structural and functional similarities, they are often labeled as CYP1A1/2 enzymes. CYP1A1/2 enzymes are not only involved in the biotransformation of clinically used drugs including paracetamol (acetaminophen) and theophylline but also involved in that of carcinogens, environmental pollutants, toxins, and natural plant products [8,9]. For example, benzo[a]pyrene (BaP) is one of the well-known polycyclic aromatic hydrocarbons which can be activated from pro-carcinogen to ultimate carcinogen by CYP1A1/2 and CYP1B1 [20]. Further, BaP is able to activate AhR, a ligand-activated transcription factor, thereby reversibly increasing the expression of CYP1A1/2 and CYP1B1 enzymes. The interest in the properties of the CYP1A enzymes stems from the preliminary results on the interaction of IP6 with other forms of CYP enzymes, which have shown that CYP1A is probably the only subfamily of human CYP enzymes which exhibits a kind of interaction with this interesting molecule [21].

Previous studies have shown that diet-derived metabolites [13] or phytochemicals [22] can modulate the metabolism of xenobiotics through the activation of AhR. Therefore, with respect to this possibility, we focused on conducting experiments with IP6 to investigate whether this phytochemical can interact with human CYP1A1/2 enzymes and participate in the AhR pathway, thus influencing drug metabolism or the activation of procarcinogens in the human organism. Until now, there were no studies evaluating IP6’s properties from the perspective of metabolism and pharmacokinetics, including drug interaction potential, in the literature.

## 2. Results

### 2.1. Difference Spectra Indicate an Interaction of IP6 with the Human Liver Microsomal Fraction (HLM) CYP

The difference spectra were obtained by subtracting the absolute spectrum of the HLM without IP6 added from the absolute spectra of the HLM with IP6. The difference spectra in the spectral region ranging from 387 to 437 nm were corrected for dilution as explained in Section 4. The spectra corresponded to the interaction of IP6 with CYP enzymes in the active site, leading to the formation of the reverse Type I (or modified type II) kind of spectral change characteristic of the absorption maximum between 415 and 435 nm, known to be associated with the interaction of substrates with weakly coordinating atoms such as the oxygen of a hydroxyl group [17,23]. The calculation of the spectral dissociation constant yielded the value of 11.2 ± 3.4 µM, indicating that IP6 interacts with cytochromes P450 (Figure 1). 

### 2.2. IP6 Does Bind to CYP1A2 According to Molecular Docking

The binding of IP6 to CYP1A2 was examined by molecular docking experiments. Since IP6 molecule bears six titratable phosphate groups, three protonation states of IP6 were considered. IP6 was set as neutral and deprotonated, bearing either a 6- or 12- negative formal charge. It is worth noting that the actual degree of ionization depends mainly on the pH of the surrounding medium but also on the ionic strength and medium dielectric constant [24]. One can expect that the most prominent protonation state of IP6 under physiological conditions corresponds to a six times negatively charged anion; however, in theory, the protonation state may change inside the active site due to local conditions (e.g., lower environment polarity with respect to water). Taking this into account and to enhance robustness of the docking experiments, we considered the three protonation states of IP6. Molecular docking was conducted with the three distinct protonation states of IP6 to examine how protonation affects its binding to the enzyme. Furthermore, the docking experiment was carried out with two distinct setups. One was centered on the CYP1A2 active site, and the other one was centered on the complete CYP1A2 enzyme to evaluate the binding of IP6 to the enzyme’s surface.

The docking experiment focused on the enzyme’s active site showed a single preferential binding pose regardless of the IP6 protonation (Figure 2). Due to its size and shape, one orientation of IP6 in the pocket situated close to the I-helix and the heme cofactor was identified. This cavity coincides with the native binding site of CYP1A2 for substrates and inhibitors, as evidenced by the crystal structure with bound α-naphthoflavone. The binding affinities were low or even positive, indicating the very low or unfavorable binding of IP6 into the CYP1A2 active site.

In contrast, when the docking grid encompassed the entire protein, different results were observed. IP6 displayed a pronounced preference for binding within four distinct molecular clusters localized on the surface of the protein (Figure 3). The most prevalent cluster (Cluster #2) was able to accommodate a total of seven poses of IP6 at various states of protonation (blue, Figure 3) and was predominantly associated with neutral IP6. This cluster was positioned in close proximity to the F-G loop, G-helix, and F-helix. This region interacts with biological membranes and represents one of the possible entrances to the active sites [25]. Partially protonated IP6, on the other hand, primarily occupied Cluster #1, with six poses at the CYP proximal side and in proximity to the K- and L-helices. Cluster #3, which had fewer poses (five), was observed across all three phosphate protonation states of IP6 and was located near the surface-facing L- and D-helices, as well as the 3–4 β sheet. Finally, the least populated molecular cluster, labeled as Cluster #4, with only three poses, was predominantly associated with ionized IP6 and found to be in close proximity to the K- and D-helices. In contrast to the docking experiment focused exclusively on the active site of CYP1A2 (Figure 2), the binding affinities of IP6 to the CYP1A2 surface were significantly lower (from −6.2 to −5.3 kcal/mol), indicating distinct preferences for IP6 binding to the enzyme’s surface, irrespective of its protonation state. Taken together, these data suggest that IP6 may bind to the enzyme’s surface; however, its affinity for the active site close to the heme is low.

### 2.3. IP6 Does Not Directly Decrease the Enzyme Activity of CYP1A1/2 in the HLM; However, It Can Increase the mRNA Level and Activity of CYP1A1/2 in Primary Human Hepatocytes (PHH) and HepG2 Cells

After finding that IP6 can interact with CYPs of the HLM (see Section 2.1) and with the CYP1A2 protein, the possibility of the presence or absence of an inhibitory effect of IP6 on the prototypical activity of CYP1A1/2 presented in the HLM was investigated. As found by docking experiments (see Section 2.2), IP6 can very weakly bind to the active site only at an acidic pH (Figure 2), with IP6 being formally neutral with all protonated phosphates. In other words, we were interested in whether the relatively weak interaction of CYP with IP6 may influence the enzyme activity. An inhibition effect of IP6 in the concentration range of 10 µM–1 mM was, however, not observed, which is in line with the possible weak interaction of IP6 with CYP 1A1/2 enzymes, which is not prominent enough to be able to influence the function of the enzyme, namely the enzyme activity (Figure 4). 

As the binding of IP6 to CYP1A2 was found to be productive—not to the active site but to the surface of the CYP1A2 protein (Figure 3, taking place in four clusters at the protein surface)—the lack of enzyme activity inhibition also shows that the active site of CYP1A2 was not affected by this mode of interaction.

Another mechanism by which IP6 can interact with drug metabolism mediated by CYPs is an interaction at the level of molecular pathways. Our focus was on functional liver cellular models including the expression and activity of CYP1A1/2 enzymes. Experiments on the viability of cells (PHH, HepG2, and HepG2-Lucia AhR reporter cells) with 3-(4,5-dimethylthiazol-2-yl)-2,5-diphenyltetrazolium bromide (MTT) have shown that cells are stable in the presence of IP6 under concentrations of 10 μM to 1 mM (Figure 5, Figure 6 and Figure 7A). The concentration range of IP6 was chosen to cover the values found in mammalian cells [26,27,28].

The effect of IP6 on CYP1A mRNA expression and activity was evaluated first in PHHs, commonly used as a “gold standard” for toxicology studies. As was mentioned in Section 1, CYP1A2 is mainly expressed in the liver, whereas CYP1A1 is mostly present in extra-hepatic tissues; thus, we focused on the mRNA expression of CYP1A2. For this purpose, PHHs obtained from five different donors were incubated for 24 h with IP6; 2,3,7,8-tetrachlordibenzo-p-dioxine (TCDD) (a prototypical CYP1A inductor [29,30]); or a combination of the two. The mRNA expression of CYP1A2 was slightly upregulated by 0.5 mM and 1 mM IP6. Likewise, 10 nM TCDD itself or in combination with 1 mM IP6 significantly upregulated (*p* < 0.05) the mRNA levels of CYP1A2 (Figure 5). Next, the enzyme activity of CYP1A1/2 [19] was measured by using the substrate 7-ethoxyresorufin (ETRR) and performing high-performance liquid chromatography (HPLC) on three samples of PHHs. An increase in the activity of CYP1A1/2 up to 126%, caused by 1 mM IP6, was observed. Different levels of induction (929–3808%) by 10 nM TCDD were observed between the three donors of PHHs, caused by the high interindividual variability of human samples. The average from these three samples was significantly increased (*p* < 0.005) compared to the control and the same as that of TCDD combined with 0.5 mM and 1 mM IP6; no further differences between these three groups were observed. 

The next step was to verify whether IP6 has the same effect in HepG2, a commonly used hepatoma cell line, considering that HepG2 has a different expression profile compared to hepatocytes, with an undetectable expression of the *CYP1A2* gene [31]. To make a conclusion about this, we determined the mRNA expression of CYP1A1, which was upregulated by 0.5 mM IP6, consistent with results from the PHHs. A slight difference between upregulation caused by TCDD and a combination of TCDD with 0.5 mM IP6 was observed (Figure 6). An increase (20%) in CYP1A1/2 activity was determined in the case of 1 mM IP6, and a significant increase was determined by using 6-formylindolol[3,2-b]carbazole (FICZ) (AhR activator), in accordance with the results derived from the experiments performed with PHHs. 

### 2.4. IP6 Does Not Activate AhR

To verify whether this upregulation (Figure 5 and Figure 6) can be directed by the AhR pathway, experiments with HepG2-Lucia AhR reporter cells were performed. Briefly, a Luciferase Reporter Assay based on AhR activation, translocation to the nucleus, the creation of a complex with aryl hydrocarbon receptor nuclear translocator (ARNT), binding to dioxin response elements, and the upregulation of target genes, including the luciferase gene, was performed. Luciferase activity was then quantified by the addition of the QUANTI-Luc substrate and luminescence measuring. No significant increase in AhR activation by IP6 was revealed. The activation of AhR by 18 µM tryptophan derivative FICZ was significant (*p* < 0.0001), and IP6 showed an antagonistic effect against this strong AhR agonist (*p* < 0.0005) in the case of 0.5 mM IP6 and (*p* < 0.0001) in the case of 1 mM IP6 (Figure 7A). As we established that IP6 can decrease the activation of AhR in combination with TCDD, the next logical step was to determine the effect of IP6 on the mRNA expression of genes related to the AhR pathway in HepG2 cells (Figure 7B). The expression of AhR itself was not affected by IP6, and similarly, TCDD did not affect AhR expression, as observed in previous publications [32,33]. The expression of the AhR repressor (AhRR) was slightly upregulated after incubation with lower concentrations of IP6 and significantly upregulated by TCDD (*p* < 0.0005), consistent with the mRNA induction of CYP1A1/2, but given previous results, this effect does not appear to be due to AhR activation.

## 3. Discussion

The main goal of this study was to assess the ability of IP6, a naturally occurring compound, to influence the metabolism of xenobiotics via interaction with cytochromes P450, as has been shown with other natural substances [34,35]. There is no report in the literature discussing IP6 in the context of food–drug interactions. The liver is the major organ where the metabolism of xenobiotics takes place; therefore, we used different hepatic models for our study—human liver microsomal fractions (HLMs), primary human hepatocytes (PHHs), and a HepG2 cell line.

The interaction of IP6 with microsomal CYPs exhibits features typical of the interaction of molecules with atoms of a low ligand field, typically, of oxygen-containing substances such as alcohols (ethanol, 1-butanol, 2-propanol) or ketones (acetone) [17,23]. The difference spectra in these cases are described as “reverse Type I” or “modified Type II”, indicating an effect of interacting substances on electron density distribution in the heme active site of kinds other than typical Type I or Type II difference spectra. 

In explaining the nature of the “reverse Type I” or “modified Type II” spectral changes, as presented in this paper, with absorption maximum at about 415 nm, the biophysical nature of these changes was often taken as a “catchall for unusual spectral changes” [18,36]. Recently, it has been noted that the relatively low ligand field strength of the ligands in question, as well as structural aspects such as the planarity of the heme, may affect the position of the absorption maxima and the amplitude of the spectral change; the suggested spectral change may be a complex manifestation of the interaction of CYPs with a class of compounds which are not acting as substrates of the CYP enzymes [18,37]. This may well be the case when the interaction of a compound with a CYP enzyme does not necessarily imply a potential drug–drug interaction [37,38].

As it was presented in the Introduction, in the current paper, the focus is on the CYP1A subfamily, which comprises two structurally similar enzymes, CYP1A1 and CYP1A2, both genes localized on chromosome 15 [39]. The mRNA expression of both genes is regulated by the AhR pathway. CYP1A2 accounts 12% of the real content of CYPs in the human liver and is involved in the metabolism of about 9% of clinically used drugs that are metabolized by CYPs. 

Docking experiments indicated that the binding of IP6 into the CYP1A2 active site is very weak and takes place only with fully protonated IP6, i.e., in highly acidic media with all six phosphate groups protonated, resulting in the molecule having a neutral formal charge. This docking experiment result is in line with the absence of an inhibitory effect of IP6 on the enzymatic activity of CYP1A/2, as the pH of the media used in our enzymatic analyses was about 7.4, preferring the partially protonated state which effectively lacks the bound IP6 in the vicinity of the heme (our attempts to locate the IP6 close to the active site at the physiological pH yielded only positive binding affinity values, i.e., the binding of IP6 to the protein’s active site was found to be improbable) (Figure 2B). With fully deprotonated IP6 (in other words, with six anionic phosphates), the situation is almost the same, with no probability of IP6 bound to the active site. On the contrary, our docking experiments revealed the possibility IP6 interacting in four regions at the surface of the protein molecule (Clusters 1, 2, 3, and 4, Figure 3A). Within each cluster, there are various poses, i.e., possible modes of IP6 orientation for either neutral (protonated, at acidic pH), partially protonated IP6 or IP6 in fully ionized form (at alkaline pH). When the possibility of binding is expressed for each of the three different ionization forms (Figure 3B), the highest probability of a binding interaction (according to values of binding affinity) can be found for partially protonated IP6 (present in Clusters 1, 2, and 3). 

Taken together, the docking experiments supported the results regarding the weak interaction of IP6 with cytochromes P450, showing weak binding affinities evidenced by spectral changes in reverse Type I characteristic spectra indicating complex interaction with compounds with low ligand fields which may not act as substrates of CYP enzymes [18,37]. The absence of an inhibitory effect on the enzyme activity of CYP1A1/2 by IP6 (at pH 7.4, partially ionized, see Section 2.3) is also in line with this reasoning, as the results of the docking studies (binding energies, see Figure 2) do not favor an interaction of substances within the active site, namely at the pH of the inhibition experiment. 

On the other hand, the experiments with cell models with IP6 (Figure 5 for hepatocytes; Figure 6 for HepG2 cells), mimicking, to some extent, the situation in the liver, showed a tendency of IP6 to modulate the metabolism of xenobiotics at the mRNA level and the activity of CYP1A enzymes. We found that the mRNA expression of CYP1A2 was upregulated by 0.5 mM and 1 mM IP6, and also, the enzymatic activity of CYP1A1/2 was increased by 1 mM IP6 in PHHs. Considering these results, we repeated the same experiments in the HepG2 cell line and observed analogous trends in the mRNA level (CYP1A1) and in the enzyme’s activity. Even though these data did not reach statistical significance, we assume that an increase in enzyme activity up to 25% can be clinically important. 

The changes in CYP1A1/2’s mRNA level and enzyme activity do not appear to be due to the AhR pathway, as has been established previously with butyrate [13]. At the basal level, IP6 did not increase AhR activation, but, interestingly, it did show an antagonistic effect against strong AhR activator FICZ in HepG2-Lucia AhR reporter cells. However, it was found that the mRNA level of AhR was not affected by the co-incubation of IP6 and TCDD in HepG2 cells, indicating that the observed antagonistic effect of IP6 toward TCDD (Figure 7A) is probably not caused by a decrease in AhR synthesis. The mRNA level of the AhR target gene (AhRR) was slightly increased by IP6, in line with the increased mRNA expression of CYP1A1/2, although significantly lower than in the case of TCDD (Figure 7B). Taken together, these AhR-dependent processes are probably not affected by IP6, and the changes in CYP1A1/2 mRNA, AhRR mRNA, and CYP1A enzyme activity are regulated in another way (as indicated in the Introduction and in the next paragraph).

In fact, it was established in an earlier publication that some structurally different, molecules of natural origin increased the mRNA expression of CYP1A through an AhR-independent mechanism [40,41]. For example, carotenoids increased the mRNA level of CYP1A through the activation of retinoid receptors (RARs), suggesting that RARs are involved in the regulation of CYP1A alone or with AhR cross-talk [40]. This effect is supported by the presence of retinoic-acid-responsive element in 5′ regulatory regions of the *CYP1A1* gene [42]. Another mechanism which could be involved in the induction of CYP1A is epigenesis. Forms of epigenetic regulation include DNA methylation, histone modifications, and noncoding RNAs. It has been shown that IP6 could be involved in mRNA-level changes in genes coding enzymes with epigenetic functions in lung tumors [43]. Some studies have also indicated the activation of histone deacetylases by IP6 or its metabolite IP3 in non-liver tissues [44,45]. However, the exact mechanism by which IP6 can upregulate CYP1A remains unclear.

The effect of IP6 on CYPs was examined in one agriculture study focused on host–plant toxic secondary metabolites [46]. The authors of this study reported a decrease in CYP activity of up to 60% in insects fed with a diet containing 1% of IP6. They concluded, based on this result, that the inhibition of CYPs can contribute to the reduced survival of plant pathogen species. Unfortunately, to date, no studies focusing on the effect of IP6 on the metabolism of xenobiotics in vertebrates, mainly in humans, have been conducted.

In conclusion, our data suggest that IP6 can slightly modulate the mRNA levels and enzyme activity of CYP1A, even though the molecular mechanism underlying this has not been revealed yet. 

Also, the potential to extrapolate the results of this study’s in vitro experiments to in vivo experiments is limited; however, thanks to the relatively weak interactions of IP6 with CYPs, mechanisms of drug–drug interactions involving IP6 are not probable.

## 4. Materials and Methods

### 4.1. Chemicals

2,3,7,8-Tetrachlordibenzo-p-dioxine (TCDD), phytic acid sodium salt hydrate, dimethyl sulfoxide (DMSO), penicillin, streptomycin, ethanol, isopropanol, TRI reagent, chloroform, methanol, acetonitrile, 3-(4,5-dimethylthiazol-2-yl)-2,5-diphenyltetrazolium bromide (MTT), bovine serum albumin, sodium pyruvate, insulin, glucagon, NADP^+^, resorufin, and Eagle’s Minimal Essential Medium (EMEM 4655) were obtained from Merck, Darmstadt, Germany. Minimum Essential Medium (MEM 31095), fetal bovine serum, glutamine, and non-essential amino acids were obtained from Gibco, Billings, MT, USA. 7-ethoxyresorufin was obtained from Lipomed, Arlesheim, Switzerland. Normocin, Zeocin, 6-formylindolol[3,2-b]carbazole (FICZ) and QUANTI–Luc were supplied by InvivoGen, Toulouse, France. Protease inhibitor cocktail tablets, EvoScript Universal cDNA, and SYBR Green reagent were obtained from Roche, Basel, Switzerland. MMLV Reverse Transcriptase Kit and TaqMan Gene Expression Assay were purchased from Invitrogen, Waltham, MA, USA. All chemicals were of the highest purity available. 

### 4.2. Human Liver Microsomal Fractions (HLMs)

Livers were obtained from multiorgan donors (5 males with an average age of 55 years and 5 females with an average age of 52 years); the use of the livers was approved by the Ethical Committee at the Faculty Hospital Olomouc. The livers were homogenized, and microsomal fractions were obtained by differential centrifugation, carried out as mentioned elsewhere [47]. The buffer for homogenization was supplemented with protease inhibitor cocktail tablets. A pool of microsomal fractions was created and stored at −80 °C. Total CYP content in the microsomal fractions was determined spectrophotometrically using difference spectra with carbon monoxide [48].

### 4.3. Spectroscopic Study of Interaction of IP6 with Human Microsomal CYP Enzymes

The absorption spectra of HLMs documenting the interaction of microsomal CYP enzymes with IP6 were acquired at room temperature using the Varian Cary 4000 UV-VIS spectrophotometer (Varian, Mulgrave, VIC, Australia) by repetitively scanning between 387 and 437 nm. Absorption difference spectra were obtained by the subtraction of a spectrum in the absence of IP6 from the individual spectra with IP6 dissolved in the K/PO_4_ buffer (see below) and corrected to dilution; HLMs were diluted to a concentration of 0.5 µM CYP using 100 mM K/PO_4_ buffer (pH 7.4). To obtain the “spectral dissociation constant” K_s_ [23], the absorption difference at 415 nm was plotted against the respective IP6 concentrations (OriginPro 2023b, Origin Lab, Northampton, MA, USA).

### 4.4. Molecular Docking of IP6 to Structure of CYP1A2

The crystal structure of the human microsomal enzyme CYP1A2 bound with α-naphthoflavone was obtained from Protein Data Bank (PDB ID: 2HI4 [49]). Missing residues were remodeled by homology modeling, which was carried out using the Modeller (ver. 10.1 [50]) tool within Chimera software 1.16 [51] according to an enzyme sequence taken from the UniProt Database (UNIPROT ID: P05177). The α-naphthoflavone was removed from the active site before a docking experiment. For the docking experiment, we considered three protonation states of IP6 due to the presence of six titratable phosphate moieties. The protonation states of IP6 were manually adjusted to match the different pH conditions: (i) acidic pH, meaning IP6 was neutral (formal charge 0); (ii) neutral pH, close to the physiological conditions, meaning IP6 was partially protonated, with each phosphate bearing a single negative charge (the IP6 formal charge was −6); (iii) alkaline pH, meaning IP6 was fully deprotonated (bearing a formal charge of −12) [24]. 

Docking experiments were carried out under two sets of conditions. In one experiment, the docking was centered around the active site, and in the other one, the docking was centered around the complete enzyme. A docking grid was set using AutoDock Tools (ver. 1.5.7) software [52], with a 34 × 14 × 30 Å grid centered at the active site of CYP1A2 next to the heme cofactor and a 68 × 54 × 74 Å grid including the whole surface of the enzyme. Polar hydrogens were added using AutoDock Tools. IP6 (in all protonation states) was docked into those two distinguishable grids using the AutoDock Vina (ver. 1.1.2.) tool [53] and software (The PyMOL Molecular Graphics System, Version 2.0 Schrödinger, LLC). We used root mean square deviation (RMSD) values to quantify the structural differences among the predicted binding poses. For all structures, we displayed their upper or lower bounds, abbreviated as u.b. or l.b., respectively.

### 4.5. Effect of IP6 on CYP1A1/2 Activity in the HLM

Enzyme activity was measured according to established protocols [19]. The incubated mixtures contained (i) 100 mM K/PO_4_ buffer (pH 7.4); (ii) 2.6 μM ETRR; (iii) an NADPH-generating system consisting of 0.8 mM NADP^+^, 6.1 mM isocitrate, 8.2 mM MgSO_4_, and 0.2 U/mL of isocitrate dehydrogenase; (iv) 35 pmol of CYPs; and (v) different concentrations of IP6 (10 μM–1 mM) as a potential inhibitor. Preincubation with IP6 was performed for 30 min; the reaction was started by adding an NADPH-generating system, performed for 15 min, and then ended by adding 200 μL of methanol. The fluorescence of the formed metabolite was measured using a Shimadzu LC-20 HPLC system (Shimadzu, Kyoto, Japan) with a LiChrospher RP-18 column (Merck, Darmstadt, Germany). The influence of IP6 on the activity of CYP1A1/2 was evaluated by plotting the respective remaining activity against IP6 concentration. All determinations were measured in triplicate in three independent experiments.

### 4.6. Human Liver Samples and Preparation of Primary Human Hepatocytes (PHHs)

Liver samples were obtained from resections performed in adult patients for medical reasons unrelated to our research program (Liv1) or from a donor when the liver was considered unsuitable for organ transplantation (Liv2). Regarding the livers from organ donors, the informed consent of the donor family was obtained by the Service de la Coordination Hospitalière (CHU Montpellier), and the protocol was approved by the French Graft Institute “Agence de la Biomedecine”. Liver resections were obtained from the Biologic Resource Center of Montpellier University Hospital (CRB-CHUM; http://www.chumontpellier.fr, accessed on 19 March 2024; Biobank ID: BB-0033-00031). Three samples of human liver were obtained from multiorgan donors, and their use was approved by the Ethics Committee of University Hospital Olomouc, Czech Republic (approval No. 119/07). The clinical characteristics of the liver donors are given in Table 1. PHHs were prepared as described previously. The hepatocytes were seeded into collagen-coated dishes at 1.7 × 10^5^ cells/cm^2^ in ISOM medium containing 2% fetal bovine serum [54]. Following 24 h of attachment, the medium was changed to ISOM without serum. 

At day 3 post-isolation, the cells were incubated for 24 h with IP6 (concentration range: 10 μM–1 mM) or with specific inductors. The enzymatic activity of CYP1A1/2 was assessed by incubating PHHs with 2.6 μM ETRR for 2 h. The supernatant was diluted with methanol (1:2) and centrifugated at 14,000 RPM at 4 °C for 10 min. Fluorescence was measured by HPLC as described above. 

### 4.7. Cell Cultures

HepG2-C3 cells (ATCCs) were cultured as recommended in MEM 31,095 or EMEM 4655 supplemented with 10% fetal bovine serum, 2 mM glutamine, 1 mM sodium pyruvate, 1% non-essential amino acids, 100 units/mL of penicillin, and 0.1 mg/mL of streptomycin in a 5% CO_2_ humidified atmosphere at 37 °C and passaged every 2–4 days. The cells were used at low passage number (<20) to ensure constitutive AhR expression. 

The enzymatic activity of CYP1A1/2 was measured via the incubation of HepG2 cells with IP6 (concentration range: 10 μM–1 mM) in a 96-well plate (Merc, Darmstadt, Germany) for 24 h. Thereafter, ETRR was added to a final concentration of 2.6 μM in a culture medium and incubated for 2 h. The supernatant was transferred into Eppendorf tubes, diluted with methanol (1:2), and centrifugated at 14,000 RPM at 4 °C for 10 min. Then, 200 μL of the supernatant was transferred into white-bottomed 96-well plates (Schoeller Instruments, Praha, Czech Republic), and fluorescence was measured using a Tecan Infinite M200 spectrophotometer (Schoeller Instruments, Praha, Czech Republic) at an excitation wavelength of 535 nm and an emission wavelength of 585 nm. The treatment of the cells followed the respective recommended procedures described in [19].

HepG2-Lucia AhR reporter cells (InvivoGen, Toulouse, France) were grown according to the manufacturer’s instructions in EMEM 4655 supplemented with 10% fetal bovine serum, 1% non-essential amino acids, 100 units/mL of penicillin, 0.1 mg/mL of streptomycin, 100 μg/mL of Normocin, and 100 μg/mL of Zeocin in a 5% CO_2_ humidified atmosphere at 37 °C and passaged every 2–4 days. 

To measure AhR activation, the cells were incubated with IP6 (concentration range: 10 μM–1 mM) in 96-well plates for 24 h. After incubation, 20 μL of the medium was transferred into white-bottomed 96-well plates, followed by the addition of 50 μL of QUANTI-Luc. Luminescence was measured using a Tecan Infinite M200 spectrophotometer.

Cell viability was measured in HepG2, HepG2-Lucia AhR reporter cells, and PHHs by the MTT test. The MTT colorimetric assay is based on the oxidoreductase activity of viable cells, which transform tetrazolium dye MTT into formazan crystals. After 24 h of treatment, the cells were washed with PBS and incubated with 5 mg/mL MTT diluted in serum-free medium (1:10) for 1 h. The mixture was aspirated, and formed formazan crystals were dissolved in DMSO/0.1% NH_3_ solution. Absorbance was spectrophotometrically measured at 540 nm.

### 4.8. RNA Isolation and PCR

Total RNA was isolated using a TRI reagent. A total of 500 ng of total RNA was reverse-transcribed using a random hexamer primer and the MMLV Reverse Transcriptase Kit or 1000 ng of total RNA was reverse-transcribed using an EvoScript Universal cDNA. Quantitative PCR was carried out using the SYBR Green reagent and a LightCycler 480 apparatus (Roche, Meylan, France) or using TaqMan Gene Expression Assays and a LightCycler 1536 Instrument (Roche, Basel, Switzerland). Subsequently, 384-well plates were pipetted using Liquid Handling Robot (EpMotion 5070, Eppendorf, Hamburg, Germany), and 1536-well plates were pipetted using Echo Liquid Handler (Labcyte, Dublin, Ireland). The amplification specificity of the SYBR Green reagent was evaluated by determining the product melting curve. Target genes were determined using primers, as shown in Table 2. The relative mRNA expression was normalized to the expression of ribosomal protein lateral stalk subunit P0 (*RPLP0*) as a housekeeping gene. The expressions of the target genes were calculated by the ΔΔCT method [55] as fold change in the treatment groups relative to the control. All treatments were measured four times.

### 4.9. Statistical Analysis

For statistical evaluation, a one-way ANOVA was carried out using GraphPad Prism 9.4 (GraphPad Software, Boston, MA, USA). Data are expressed as means ± SD. The number of experiments and replicates are given in each figure legend. Differences were regarded as statistically significant when the *p* value was lower than 0.05.

## Figures and Tables

**Figure 1 ijms-25-03610-f001:**
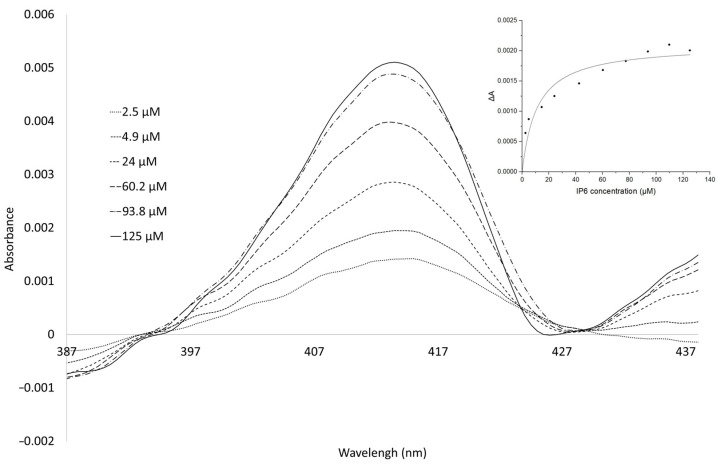
Spectral determination of the interaction between IP6 and CYPs in the HLM. The concentration of CYPs in the microsomal preparation was 0.5 µM in 100 mM K/PO_4_ buffer (pH 7.4); the IP6 concentrations were 2.5–125 µM. Insert: a plot of the spectral difference (ΔA) at about 415 nm versus IP6 concentration for the determination of the binding constant (Ks).

**Figure 2 ijms-25-03610-f002:**
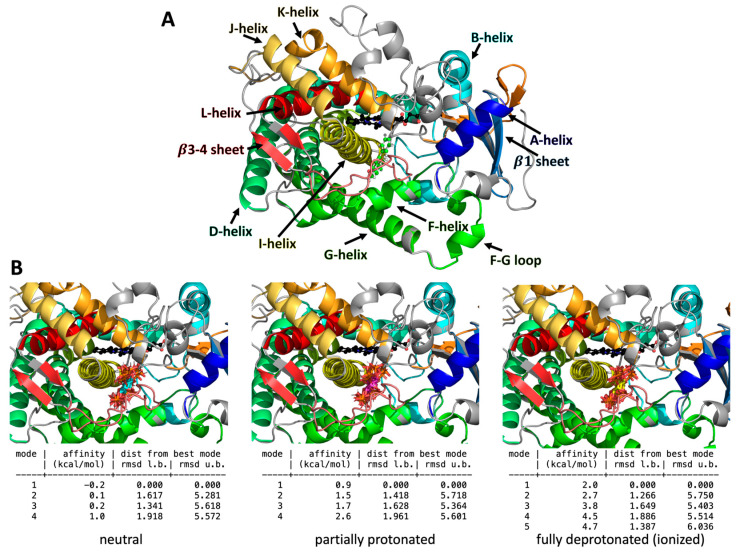
(**A**) Representative structure of CYP1A2 with bound α-naphthoflavone, shown as the green spheres to the right of the I-helix, and (**B**) predicted docking poses (displayed as four or five binding modes) of IP6 in three protonation states docked directly within the active site of CYP1A2. In (**B**), IP6 is shown as sticks with the following coloring pattern: cyan for neutral, magenta for partially protonated, and yellow for ionized; their binding affinities and root mean square deviation (RMSD) values from the best pose (mode) are shown as references. Only a single orientation of IP6 was found in all three protonation states, with IP6 being located below the heme cofactor (black spheres) and to the right of the I-helix (shown in yellow).

**Figure 3 ijms-25-03610-f003:**
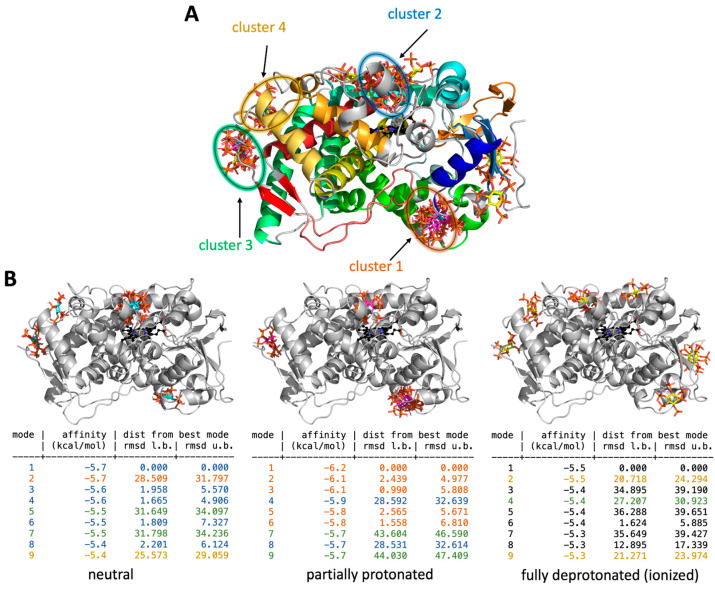
(**A**,**B**) Molecular clusters of IP6 on the enzyme surface identified by the docking experiment. (**A**) Docking poses (modes) of IP6 in three protonation states docked to the entire enzyme (IP6 is shown as sticks with the following coloring pattern: cyan for neutral, magenta for partially protonated, and yellow for ionized states); their binding affinities and RMSD values from the best pose are used as references. (**B**) The cluster color schemes of the poses from panels (**A**,**B**) are organized as follows: Cluster #1 in orange, Cluster #2 in blue, Cluster #3 in green, and Cluster #4 in yellow.

**Figure 4 ijms-25-03610-f004:**
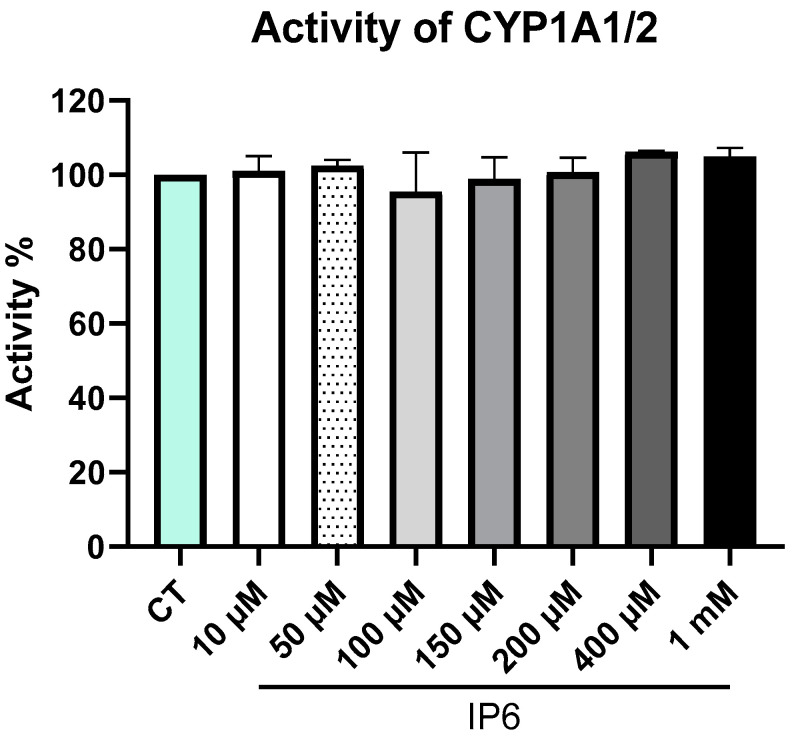
Effects of IP6 on catalytic activity of CYP1A1/2 in the HLM. All experiments were performed in triplicate, all data were normalized to the control (CT—mint-colored bar), and each bar represents the mean ± SD of three independent experiments.

**Figure 5 ijms-25-03610-f005:**
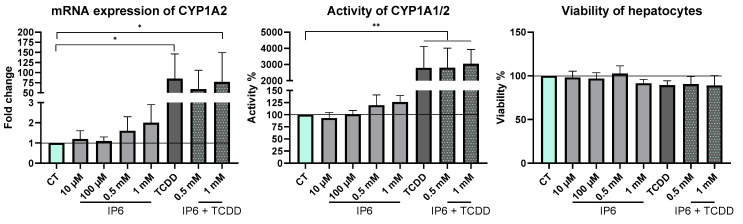
Effect of IP6 on the mRNA level and activity of CYP1A and viability in PHHs after 24 h of treatment. We used 10 nM TCDD as a positive control. All experiments were performed in triplicate, all data were normalized to the control (CT—mint bars), and each bar represents the mean ± SD of independent experiments. Five samples of human liver from donors were used to determine viability and mRNA expressions, three samples of which were used for the assessment of enzyme activity. (* *p* < 0.05; ** *p* < 0.005).

**Figure 6 ijms-25-03610-f006:**
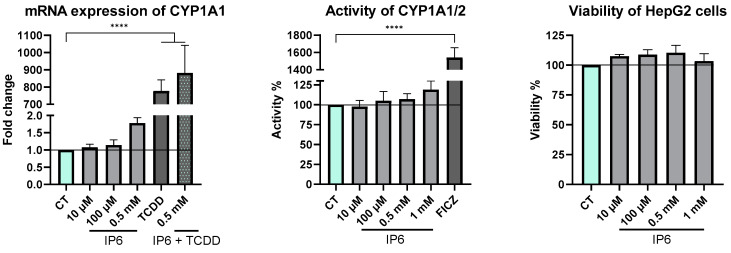
Effect of IP6 on the mRNA level and activity of CYP1A and viability in the HepG2 cell line after 24 h of treatment. We used 10 nM TCDD and 18 µM FICZ as positive controls. All experiments were performed in triplicate, all data were normalized to the control (CT—mint bars), and each bar represents the mean ± SD of three independent experiments. (**** *p* < 0.0001).

**Figure 7 ijms-25-03610-f007:**
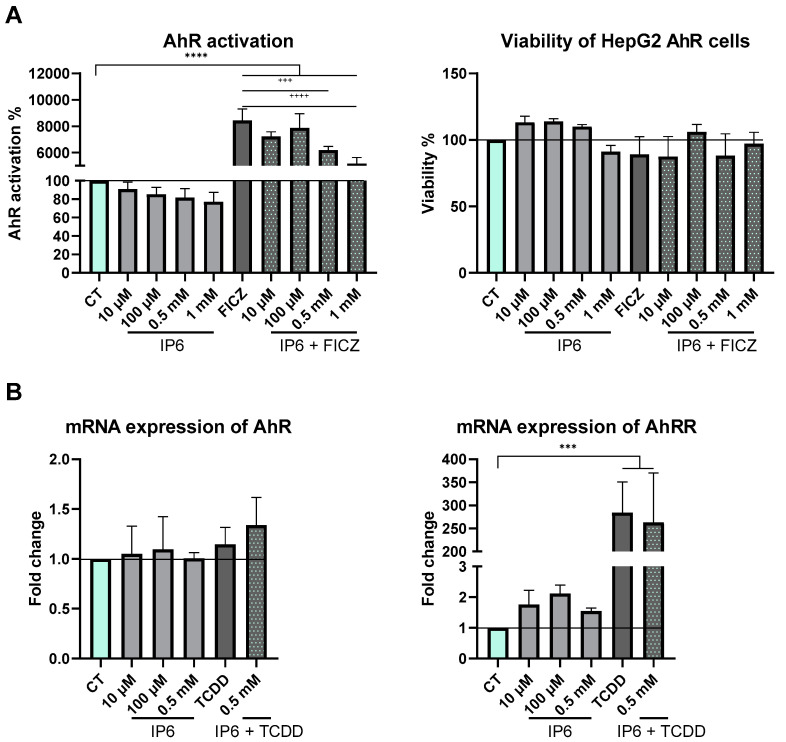
(**A**) Effect of IP6 on the AhR activation and viability in the HepG2-Lucia AhR reporter cell line after 24 h of treatment. We used 18 µM FICZ as a positive control. (**B**) Effect of IP6 and 10 nM TCDD on gene expression in HepG2 cells. All experiments were performed in triplicate, all data were normalized to the control (CT—mint bars), and each bar represents the mean ± SD of three independent experiments. (*** *p* < 0.0005; **** *p* < 0.0001; ^+++^ *p* < 0.0005; and ^++++^ *p* < 0.0001 compared to FICZ).

**Table 1 ijms-25-03610-t001:** Clinical characteristics of the liver donors.

Liver Identification	Gender	Age	Pathology
Liv1	Man	58	Hepatocellular carcinoma
Liv2	Man	80	Organ donor
LH93	Man	56	Organ donor
LH94	Woman	78	Organ donor
LH95	Woman	79	Organ donor

**Table 2 ijms-25-03610-t002:** Primer sequences.

Gene	Forward Primer	Reverse Primer
*RPLP0*	TCGACAATGGCAGCATCTAC	GCCTTGACCTTTTCAGCAAG
*AhR*	GTCGTCTAAGGTGTCTGCTGGA	CGCAAACAAAGCCAACTGAGGTG
*AhRR*	TGACCTTGTCCTTGACCC	CCATCCTCACTGTGCTTTC
*CYP1A1*	TCCGGGACATCACAGACAGC	ACCCTGGGGTTCATCACCAA
*CYP1A2*	CATCCCCCACAGCACAACAA	TCCCACTTGGCCAGGACTTC

## Data Availability

The raw data supporting the conclusions of this manuscript will be made available by the authors, without reservation, to any qualified researcher.

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
