# Peer review of "On the Possible Effect of Phytic Acid (Myo-Inositol Hexaphosphoric Acid, IP6) on Cytochromes P450 and Systems of Xenobiotic Metabolism in Different Hepatic Models"

_ijms, 2024, doi:10.3390/ijms25073610_

Round 1

Reviewer 1 Report

Comments and Suggestions for Authors

The study conducted by Frýbortová et al. provides valuable insights into the interaction between phytic acid (IP6) and cytochrome P450 (CYP) enzyme CYP1A, crucial player in drug metabolism. The investigation is particularly significant considering the increasing interest in natural compounds and their potential effects on human physiology, including their interactions with drug metabolism pathways. The authors employed a comprehensive approach, utilizing human liver microsomes, primary human hepatocytes, the HepG2 cell line, and molecular docking techniques to elucidate the interaction between IP6 and CYP1A1/2 enzyme. The use of multiple experimental models strengthens the reliability and relevance of the findings. The results are clearly presented, detailing the findings from each experimental approach. I believe the work makes a valuable contribution to the field. However, there are few considerations that need to be addressed:

1. It would be beneficial for the authors to discuss the limitations of the study, such as the use of in vitro models and the extrapolation of findings to in vivo settings. Addressing these limitations would provide a more balanced perspective on the implications of the results.

2. Better resolution images for Figure 1, 2 and 3

3. For HepG2 experiments, data for 1uM IP6 seems to be missing for mRNA expression, can the authors clarify on this part?

Minor comments:

1. Phytic acid has been described with different names in the abstract and introduction section, make sure there is consistency

2. Make sure all abbreviations have been defined in their respective first appearance in the text. For example, no definition for HLM, PHH.

Comments on the Quality of English Language

There are a few minor grammatical errors and spelling errors throughout the manuscript

Reviewer 2 Report

Comments and Suggestions for Authors

This study deals with the effect of phytic acid on CYP1A2 using HepG2 cells. Drug-drug interactions with phytic acid were found to be less possibility with respect to CYP1A2.  The study using HepG2 cells also suggests that phytic acid has little effect on the induction of CYP1A2 by TCDD or on its activity. In addition, the effects on the expression of AHRR the gene also regulated through AhR are examined using not only TCDD but also the endogenous AhR ligand FICZ. These results suggest that phytic acid has little effect on CYP1A2-mediated drug oxidation. This study is well designed. The results sound.

Comments on this paper are as follows:

Major concerns: 

1. Please explain how the authors decided the concentration of phytic acid employed in the study? What is the normal blood concentration range of phytic acid, which may also be endogenous? What is the range of blood concentrations of phytic acid when taken in the diet or as a supplement? Using your own data or appropriate references, justify the concentrations of phytic acid employed in this study. If refer to previous work, please add new reference(s) if needed.

2. In the inhibition study on CYP1A2 activity by phytic acid,

Did the authors determine the Km value of CYP1A2 activity? Did you choose the Km to study the conditions for inhibition or activation? If so, provide the Km value. If necessary, please cite the appropriate references.

3. Similarly, provide the rationale for selecting the TCDD and FICZ concentrations employed. Also please cite the appropriate literatures.

4. This study was conducted with a focus on CYP1A2, which is a good point. On the other hand, there are other CYP isoforms in the human liver, including CYP3A4. Is there a need to study the effect of phytic acid on P450s other than CYP1A2? Do the authors have examples of previous studies? Or, if available in a previous report, cite and discuss this point in the discussion section.

Reviewer 3 Report

Comments and Suggestions for Authors

In the current study, the authors studied the interaction between P450 enzyme, mainly CYP1A enzymes, and IP6, a natural compound. The authors found that IP6 can directly bind with CYP1A2 enzyme and regulate the mRNA expression of CYP1A2. However, the interaction is defined as weak and support the conclusion that IP6 may not likely to cause drug-drug interactions.

I have few suggestions here which I hope can help improving the quality of the manuscript.

In the introduction section, I would suggest adding more information about currently identified mechanism in term of CYP450 regulation. At least, there are direct mechanisms, such as direct binding to P450 enzymes like IP6, and indirect mechanisms, such as transcriptional regulation mentioned here.

Better justification of why specifically look at CYP1A family is needed here as CYP1A only account for around 7-9% of all P450 mediated drug metabolism.

In the cell experiments, the concentration of IP6 used ranged from 10 uM to 1 mM. I wonder what the range of IP6 concentration in human will be. Dose the in vitro experiments have any correlation with in vivo prediction?

If the IP6 can affect enzymatic activity of CYP1A2, I would suggest the authors perform IP6-drug interaction assays to see if IP6 can alter drug activity/toxicity through regualtion of CYP1A2. APAP would be an ideal model drug to use and the outcome can be measured by generation of ROS or cell death.

Transcriptional regulation of P450 enzymes are not limited to transcription factors. I would suggest the authors add some discussion about other mechanisms which are potentially responsible for the altered CYP1A2 mRNA levels in the discussion part.

  Comments on the Quality of English Language

Some grammer issues were found (example: line 35-36). Proofread is suggested.

The use of abbreviations needs to be revised. If the full name only appeared once in the whole content, no abbreviation is needed.
